# Grounded Video Situation Recognition

**Zeeshan Khan**    **C. V. Jawahar**    **Makarand Tapaswi**
CVIT, IIIT Hyderabad
https://zeeshank95.github.io/grvidsitu

## Abstract

Dense video understanding requires answering several questions such as *who is doing what to whom, with what, how, why, and where*. Recently, Video Situation Recognition (VidSitu) is framed as a task for structured prediction of multiple events, their relationships, and actions and various verb-role pairs attached to descriptive entities. This task poses several challenges in identifying, disambiguating, and co-referencing entities across multiple verb-role pairs, but also faces some challenges of evaluation. In this work, we propose the addition of spatio-temporal grounding as an essential component of the structured prediction task in a weakly supervised setting, and present a novel three stage Transformer model, *VideoWhisperer*, that is empowered to make joint predictions. In stage one, we learn contextualised embeddings for video features in parallel with key objects that appear in the video clips to enable fine-grained spatio-temporal reasoning. The second stage sees verb-role queries attend and pool information from object embeddings, localising *answers* to questions posed about the action. The final stage generates these answers as captions to describe each verb-role pair present in the video. Our model operates on a group of events (clips) simultaneously and predicts verbs, verb-role pairs, their nouns, and their grounding on-the-fly. When evaluated on a grounding-augmented version of the VidSitu dataset, we observe a large improvement in entity captioning accuracy, as well as the ability to localize verb-roles without grounding annotations at training time.

## 1   Introduction

At the end of *The Dark Knight*, we see a short intense sequencethat involves Harvey Dent toss a coin while holding a gun followed by sudden action. Holistic understanding of such a video sequence, especially one that involves multiple people, requires predicting more than the action label (*what* verb). For example, we may wish to answer questions such as *who* performed the action (agent), *why* they are doing it (purpose / goal), *how* are they doing it (manner), *where* are they doing it (location), and even *what happens after* (multi-event understanding). While humans are able to perceive the situation and are good at answering such questions, many works often focus on building tools for doing single tasks, *e.g.* predicting actions [8] or detecting objects [2, 4] or image/video captioning [18, 28]. We are interested in assessing how some of these advances can be combined for a holistic understanding of video clips.

A recent and audacious step towards this goal is the work by Sadhu *et al.* [27]. They propose Video Situation Recognition (VidSitu), a structured prediction task over five short clips consisting of three sub-problems: (i) recognizing the salient actions in the short clips; (ii) predicting roles and their entities that are part of this action; and (iii) modelling simple event relations such as enable or cause. Similar to the predecessor image situation recognition (imSitu [39]), VidSitu is annotated using Semantic Role Labelling (SRL) [21]. A video (say 10s) is divided into multiple small events (~2s) and each event is associated with a salient action verb (*e.g. hit*). Each verb has a fixed set of roles or arguments, *e.g. agent-Arg0*, *patient-Arg1*, *tool-Arg2*, *location-ArgM(Location)*, *manner-*

*ArgM(manner)*, *etc.*, and each role is annotated with a free form text caption, *e.g. agent: Blonde Woman*, as illustrated in Fig. 1.

**Grounded VidSitu.** VidSitu poses various challenges: long-tailed distribution of both verbs and text phrases, disambiguating the roles, overcoming semantic role-noun pair sparsity, and co-referencing of entities in the entire video. Moreover, there is ambiguity in text phrases that refer to the same unique entity (*e.g.* "man in white shirt" or "man with brown hair"). A model may fail to understand which attributes are important and may bias towards a specific caption (or pattern like shirt color), given the long-tailed distribution. This is exacerbated when multiple entities (*e.g. agent* and *patient*) have similar attributes and the model predicts the same caption for them (see Fig. 1). To remove biases of the captioning module and gauge the model's ability to identify the role, we propose *Grounded Video Situation Recognition* (GVSR) - an extension of the VidSitu task to include spatio-temporal grounding. In addition to predicting the captions for the role-entity pairs, we now expect the structured output to contain spatio-temporal localization, currently posed as a weakly-supervised task.

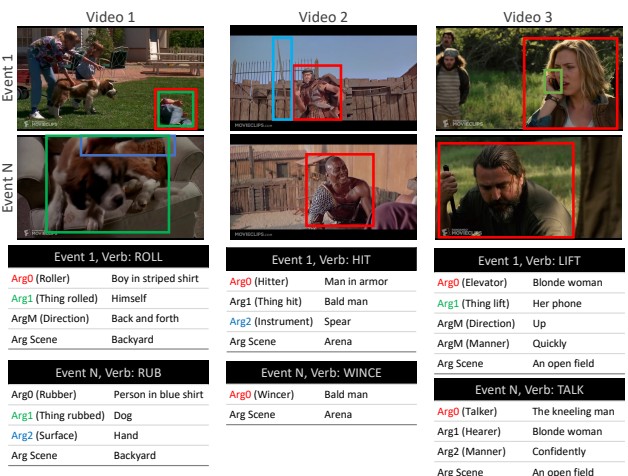

Figure 1: **Overview of GVSR:** Given a video consisting of multiple events, GVSR requires recognising the action verbs, their corresponding roles, and localising them in the spatio-temporal domain. This is a challenging task as it requires to disambiguate between several roles that the same entity may take in different events, *e.g.* in Video 2 the *bald man* is a *patient* in event 1, but an *agent* in event N. Moreover, the entities present in multiple events are co-referenced in all such events. Colored arguments are grounded in the image with bounding boxes (figure best seen in colour).

**Joint structured prediction.** Previous works [27, 37] modeled the VidSitu tasks separately, *e.g.* the ground-truth verb is fed to the SRL task. This setup does *not* allow for situation recognition on a new video clip without manual intervention. Instead, in this work, we focus on solving three tasks jointly: (i) verb classification; (ii) SRL; and (iii) Grounding for SRL. We ignore the original event relation prediction task in this work, as this can be performed later in a decoupled manner similar to [27].

We propose *VideoWhisperer*, a new three-stage transformer architecture that enables video understanding at a global level through self-attention across all video clips, and generates predictions for the above three tasks at an event level through localised event-role representations. In the first stage, we use a Transformer encoder to align and contextualise 2D object features in addition to event-level video features. These rich features are essential for grounded situation recognition, and are used to predict both the verb-role pairs and entities. In the second stage, a Transformer decoder models the role as a query, and applies cross-attention to find the best elements from the contextualised object features, also enabling visual grounding. Finally, in stage three, we generate the captions for each role entity. The three-stage network disentangles the three tasks and allows for end-to-end training.

**Contributions summary.** (i) We present a new framework that combines grounding with SRL for end-to-end Grounded Video Situation Recognition (GVSR). We will release the grounding annotations and also include them in the evaluation benchmark. (ii) We design a new three-stage transformer architecture for joint verb prediction, semantic-role labelling through caption generation, and weakly-supervised grounding of visual entities. (iii) We propose role prediction and use role queries contextualised by video embeddings for SRL, circumventing the requirement of ground-truth verbs or roles, enabling end-to-end GVSR. (iv) At the encoder, we combine object features with video features and highlight multiple advantages enabling weakly-supervised grounding and improving the quality of SRL captions leading to a 22 points jump in CIDEr score in comparison to a video-only baseline [27]. (v) Finally, we present extensive ablation experiments to analyze our model. Our model achieves the state-of-the-art results on the VidSitu benchmark.

## 2    Related Work

**Image Situation Recognition.** Situation Recognition in images was first proposed by [10] where they created datasets to understand actions along with localisation of objects and people. Another line of work, imSitu [39] proposed situation recognition via semantic role labelling by leveraging linguistic frameworks, FrameNet [3] and WordNet [19] to formalize situations in the form of verb-role-noun triplets. Recently, grounding has been incorporated with image situation recognition [23] to add a level of understanding for the predicted SRL. Situation recognition requires global understanding of the entire scene, where the verbs, roles and nouns interact with each other to predict a coherent output. Therefore several approaches used CRF [39], LSTMs [23] and Graph neural networks [14] to model the global dependencies among verb and roles. Recently various Transformer [32] based methods have been proposed that claim large performance improvements [6, 7, 35].

**Video Situation Recognition.** Recently, imSitu was extended to videos as VidSitu [27], a large scale video dataset based on short movie clips spanning multiple events. Compared to image situation recognition, VidSRL not only requires understanding the action and the entities involved in a single frame, but also needs to coherently understand the entire video while predicting event-level verb-SRLs and co-referencing the entities participating across events. Sadhu *et al*. [27] propose to use standard video backbones for feature extraction followed by multiple but separate Transformers to model all the tasks individually, using ground-truth of previous the task to model the next. A concurrent work to this submission, [37], proposes to improve upon the video features by pretraining the low-level video backbone using contrastive learning objectives, and pretrain the high-level video contextualiser using event mask prediction tasks resulting in large performance improvements on SRL. Our goals are different from the above two works, we propose to learn and predict all three tasks simultaneously. To achieve this, we predict verb-role pairs on the fly and design a new role query contextualised by video embeddings to model SRL. This eliminates the need for ground-truth verbs and enables end-to-end situation recognition in videos. We also propose to learn contextualised object and video features enabling weakly-supervised grounding for SRL, which was not supported by previous works.

**Video Understanding.** Video understanding is a broad area of research, dominantly involving tasks like action recognition [5, 8, 9, 29, 34, 36], localisation [16, 17], object grounding [26, 38], question answering [31, 40], video captioning [25], and spatio-temporal detection [9, 30]. These tasks involve visual temporal understanding in a sparse uni-dimensional way. In contrast, GVSR involves a hierarchy of tasks, coming together to provide a fixed structure, enabling dense situation recognition. The proposed task requires global video understanding through event level predictions and fine-grained details to recognise all the entities involved, the roles they play, and simultaneously ground them. Note that our work on grounding is different from classical spatio-temporal video grounding [41, 38] or referring expressions based segmentation [11] as they require a text query as input. In our case, both the text and the bounding box (grounding) are predicted jointly by the model.

## 3    VideoWhisperer for Grounded Video Situation Recognition

We now present the details of our three stage Transformer model, *VideoWhisperer*. A visual overview is presented in Fig. 2. For brevity, we request the reader to refer to [32] for now popular details of self- and cross-attention layers used in Transformer encoders and decoders.

**Preliminaries.** Given a video $V$ consisting of several short events $\mathcal{E} = \{e_i\}$, the complete situation in $V$, is characterised by 3 tasks. (i) Verb classification, requires predicting the action label $v_i$ associated with each event $e_i$; (ii) Semantic role labelling (SRL), involves guessing the nouns (captions) $\mathcal{C}_i = \{C_{ik}\}$ for various roles $\mathcal{R}_i = \{r | r \in \mathcal{P}(v_i) \forall r \in \mathcal{R}\}$ associated with the verb $v_i$. $\mathcal{P}$ is a mapping function from verbs to a set of roles based on VidSitu (extended PropBank [21]) and $\mathcal{R}$ is the set of all roles); and (iii) Spatio-temporal Grounding of each visual role-noun prediction $C_{ij}$ is formulated as selecting one among several bounding box proposals $\mathcal{B}$ obtained from sub-sampled keyframes of the video. We evaluate this against ground-truth annotations done at a keyframe level.

### 3.1    Contextualised Video and Object Features (Stage 1)

GVSR is a challenging task, that requires to coherently model spatio-temporal information to understand the salient action, determine the semantic role-noun pairs involved with the action, and simultaneously localise them. Different from previous works that operate only on event level

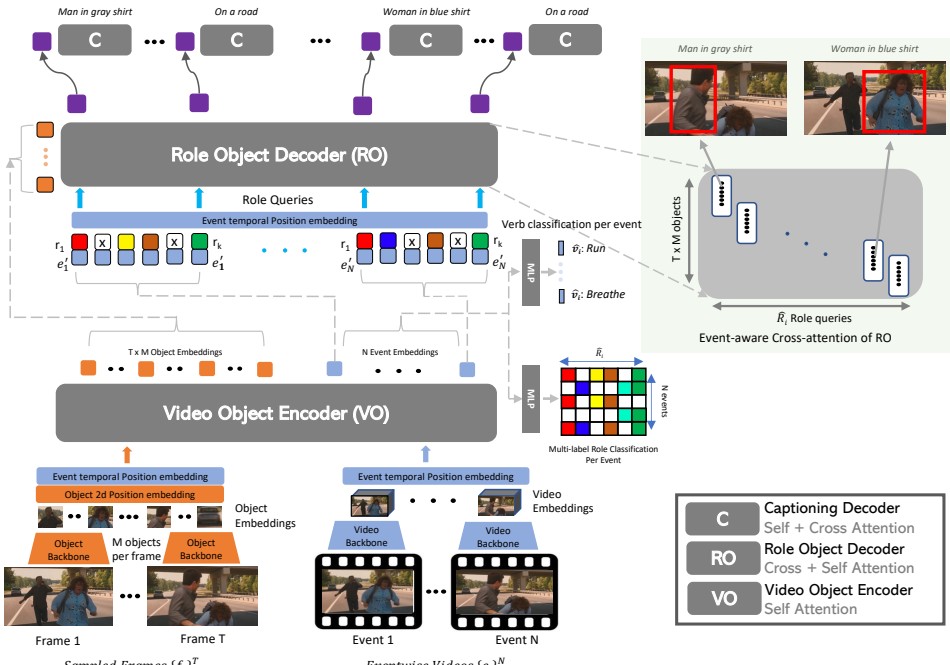

Figure 2: **VideoWhisperer**: We present a new 3-stage Transformer for GVSR. Stage-1 learns the contextualised object and event embeddings through a video-object Transformer encoder (VO), that is used to predict the verb-role pairs for each event. Stage-2 models all the predicted roles by creating role queries contextualised by event embeddings, and attends to all the object proposals through a role-object Transformer decoder (RO) to find the best entity that represents a role. The output embeddings are fed to captioning Transformer decoder (C) to generate captions for each role. Transformer RO's cross-attention ranks all the object proposals enabling localization for each role.

video features, we propose to model both the event and object level features simultaneously. We use a pretrained video backbone $\phi_{\text{vid}}$ to extract event level video embeddings $\mathbf{x}_i^e = \phi_{\text{vid}}(e_i)$. For representing objects, we subsample frames $\mathcal{F} = \{f_t\}_{t=1}^T$ from the entire video $V$. We use a pretrained object detector $\phi_{\text{obj}}$ and extract top $M$ object proposals from every frame. The box locations (along with timestamp) and corresponding features are

$$\mathcal{B} = \{b_{mt}\}, m = [1, \ldots, M], t = [1, \ldots, T], \quad \text{and} \quad \{\mathbf{x}_{mt}^o\}_{m=1}^M = \phi_{\text{obj}}(f_t) \quad \text{respectively.} \quad (1)$$

The subset of frames associated with an event $e_i$ are computed based on the event's timestamps,

$$\mathcal{F}_i = \{f_t | e_i^{\text{start}} \leq t \leq e_i^{\text{end}}\}. \quad (2)$$

Specifically, at a sampling rate of 1fps, video $V$ of 10s, and events $e_i$ of 2s each, we associate 3 frames with each event such that the border frames are shared. We can extend this association to all object proposals based on the frame in which they appear and denote this as $\mathcal{B}_i$.

**Video-Object Transformer Encoder (VO).** Since the object and video embeddings come from different spaces, we align and contextualise them with a Transformer encoder [32]. Event-level position embeddings $\text{PE}_i$ are added to both representations, event $\mathbf{x}_i^e$ and object $\mathbf{x}_{mt}^o$ ($t \in \mathcal{F}_i$). In addition, 2D object position embeddings $\text{PE}_{mt}$ are added to object embeddings $\mathbf{x}_{mt}^o$. Together, they help capture spatio-temporal information. The object and video tokens are passed through multiple self-attention layers to produce contextualised event and object embeddings:

$$[\ldots, \mathbf{o}_{mt}', \ldots, \mathbf{e}_i', \ldots] = \text{Transformer}_{\text{VO}}([\ldots, \mathbf{x}_{mt}^o + \text{PE}_i + \text{PE}_{mt}, \ldots, \mathbf{x}_i^e + \text{PE}_i, \ldots]). \quad (3)$$

**Verb and role classification.** Each contextualised event embedding $\mathbf{e}_i'$ is not only empowered to combine information across neighboring events but also focus on key objects that may be relevant. We predict the action label for each event by passing them through a 1-hidden layer MLP,

$$\hat{v}_i = \text{MLP}_e(\mathbf{e}_i'). \quad (4)$$

Each verb is associated with a fixed set of roles based on the mapping $\mathcal{P}(\cdot)$. This prior information is required to model the SRL task. Previous works [27, 37] use ground-truth verbs to model SRL and predict both the roles and their corresponding entities. While this setup allows for task specific modelling, it is not practical in the context of end-to-end video situation recognition. To enable GVSR, we predict the relevant roles for each event circumventing the need for ground-truth verbs and mapped roles. Again, we exploit the contextualised event embeddings and pass them through a role-prediction MLP and perform *multi-label* role classification. Essentially, we estimate the roles associated with an event as

$$\hat{\mathcal{R}}_i = \{r | \sigma(\mathrm{MLP}_r(\mathbf{e}'_i)) > \theta_{\mathrm{role}}\}, \tag{5}$$

where $\sigma(\cdot)$ is the sigmoid function and $\theta_{\mathrm{role}}$ is a common threshold across all roles (typically set to 0.5). Armed with verb and role predictions, $\hat{v}_i$ and $\hat{\mathcal{R}}_i$, we now look at localising the role-noun pairs and generating the SRL captions.

## 3.2 Semantic Role Labelling with Grounding (Stage 2, 3)

A major challenge in SRL is to disambiguate roles, as the same object (person) may take on different roles in the longer video $V$. For example, if two people are conversing, the *agent* and *patient* roles will switch between *speaker* and *listener* over the course of the video. Another challenge is to generate descriptive and distinctive captions for each role such that they refer to a specific entity. We propose to use learnable role embeddings $\{\mathbf{r}_{ik}\}_{k=1}^{|\mathcal{R}_i|}$ which are capable of learning distinctive role representations. As mentioned earlier, roles such as *agent*, *patient*, *tool*, *location*, *manner*, *etc.* ask further questions about the salient action.

**Creating role queries.** Each role gets updated by the verb. For example, for an action *jump*, the *agent* would be referred to as the *jumper*. We strengthen the role embeddings by adding the contextualised event embeddings to each role, instead of encoding ground-truth verbs. This eliminates the dependency on the ground-truth verb-role pairs, and enables end-to-end GVSR. Similar to the first stage (VO), we also add event-level temporal positional embeddings to obtain role *query* vectors

$$\mathbf{q}_{ik} = \mathbf{r}_k + \mathbf{e}'_i + \mathrm{PE}_i. \tag{6}$$

Depending on the setting, $k$ can span all roles $\mathcal{R}$, ground-truth roles $\mathcal{R}_i$ or predicted roles $\hat{\mathcal{R}}_i$.

**Role-Object Transformer Decoder (RO).** It is hard to achieve rich captions while using features learned for action recognition. Different from prior works [27, 37], we use fine-grained object level representations instead of relying on event-based video features. We now describe the stage two of our VideoWhisperer model, the Transformer decoder for SRL.

Our Transformer decoder uses semantic roles as queries and object proposal representations as keys and values. Through the cross-attention layer, the event-aware role query attends to contextualised object embeddings and finds the best objects that represent each role. We incorporate an event-based attention mask, that limits the roles corresponding to an event to search for objects localised in the same event, while masking out objects from other events. Cross-attention captures local event-level role-object interactions while the self-attention captures the global video level understanding allowing event roles to share information with each other.

We formulate event-aware cross-attention as follows. We first define the query, key, and value tokens fed to the cross-attention layer as

$$\mathbf{q}'_{ik} = W_Q \mathbf{q}_{ik}, \quad \mathbf{k}'_{mt} = W_K \mathbf{o}'_{mt}, \quad \text{and} \quad \mathbf{v}'_{mt} = W_V \mathbf{o}'_{mt}. \tag{7}$$

Here, $W_{[Q|K|V]}$ are learnable linear layers. Next, we apply a mask while computing cross-attention to obtain contextualised role embeddings as

$$\mathbf{r}'_{ik} = \sum_{mt} \alpha_{mt} \mathbf{v}'_{mt}, \quad \text{where} \quad \alpha_{mt} = \mathrm{softmax}_{mt}(\langle \mathbf{q}'_{ik}, \mathbf{k}'_{mt} \rangle \cdot \mathbb{1}(f_t \in \mathcal{F}_i)), \tag{8}$$

where $\langle \cdot, \cdot \rangle$ is an inner product and $\mathbb{1}(\cdot)$ is an indicator function with value 1 when true and $-\infty$ otherwise to ensure that the cross-attention is applied only to the boxes $\mathcal{B}_i$, whose frames $f_t$ appear within the same event $e_i$.

After multiple layers of cross- and self-attention, the role query extracts objects that best represent the entities for each role.

$$[\ldots, \mathbf{z}_{ik}, \ldots] = \mathrm{Transformer}_{\mathrm{RO}}([\ldots, \mathbf{q}_{ik}, \ldots; \ldots, \mathbf{o}'_{mt}, \ldots]). \tag{9}$$

**Captioning Transformer Decoder (C).** The final stage of our model is a caption generation module. Specifically, we use another Transformer decoder [32] whose input context is the output role embedding $\mathbf{z}_{ik}$ from the previous stage and unroll predictions in an autoregressive manner.

$$\hat{C}_{ik} = \text{Transformer}_C(\mathbf{z}_{ik}) \,. \tag{10}$$

The role-object decoder in stage 2 shares all the necessary information through self-attention, and allows us to generate the captions for all the roles in parallel; while [27, 37] generate captions sequentially , *i.e.* for a given event, the caption for role $k$ is decoded only after the caption for role $k-1$. This makes VideoWhisperer efficient with a wall-clock runtime of 0.4s for inference on a 10s video, while the baseline [27] requires 0.94 seconds.

**Grounded Semantic Role Labelling.** The entire model is designed in a way to naturally provide SRL with grounding in a weakly-supervised way, without the need for ground-truth bounding boxes during training. Cross-attention through the Transformer decoder RO scores and ranks all the objects based on the role-object relevance at every layer. We extract the cross-attention scores $\alpha_{mt}$ for each role $k$ and event $e_i$ from the final layer of Transformer$_{\text{RO}}$, and identify the highest scoring box and the corresponding timestep as

$$\hat{b}_m^*, \hat{b}_t^* = \arg\max_{m,t} \alpha_{mt} \,. \tag{11}$$

### 3.3 Training and Inference

**Training.** VideoWhisperer can be trained in an end-to-end fashion, with three losses. The first two losses, CrossEntropy and BinaryCrossEntropy, are tapped from the contextualis ed event embeddings and primarily impact the Video-Object Transformer encoder

$$L_i^{\text{verb}} = CE(\hat{v}_i, v_i) \quad \text{and} \quad L_i^{\text{role}} = \sum_{r \in \mathcal{R}_i} BCE(r \in \hat{\mathcal{R}}_i, r \in \mathcal{R}_i) \,. \tag{12}$$

The final component is derived from the ground-truth captions and helps produce meaningful SRL outputs. This is also the source of weak supervision for the grounding task,

$$L_{ik}^{\text{caption}} = \sum_w CE(\hat{C}_{ik}^w, C_{ik}^w) \,, \tag{13}$$

where the loss is applied in an autoregressive manner to each predicted word $w$. The combined loss for any training video $V$ is given by

$$\mathcal{L} = \sum_i L_i^{\text{verb}} + \sum_i L_i^{\text{role}} + \sum_{ik} L_{ik}^{\text{caption}} \,. \tag{14}$$

**Inference.** At test time, we split the video $V$ into similar events $e_i$ and predict verbs $\hat{v}_i$ and roles $\hat{\mathcal{R}}_i$ for the same. Here, we have two options: (i) we can use the predicted verb and obtain the corresponding roles using a ground-truth mapping between verbs and roles $\mathcal{P}(\hat{v}_i)$, or (ii) only predict captions for the predicted roles $\hat{\mathcal{R}}_i$. We show the impact of these design choices through experiments.

## 4 Experiments

We evaluate our model in two main settings. (i) This setup mimics VidSitu [27], where tasks are evaluated separately. We primarily focus on (a) Verb prediction, (b) SRL and (c) Grounded SRL. This setting uses ground-truth verb-role pairs for modelling (b) and (c). (ii) End-to-end GVSR, where all the three tasks are modelled together without using ground truth verb-roles.

**Dataset.** We evaluate our model on the VidSitu [27] dataset that consists of 29k videos (23.6k train, 1.3k val, and others in task-specific test sets) collected from a diverse set of 3k movies. All videos are truncated to 10 seconds, have 5 events spanning 2 seconds each and are tagged with verb and SRL annotations. There are a total of 1560 verb classes and each verb is associated with a fixed set of roles among 11 possible options, however not all are used for evaluation due to noisy annotations (we follow the protocol by [27]). For each role the corresponding value is a free-form caption.

**Metrics.** For verb prediction, we report Acc@K, *i.e.* event accuracy considering 10 ground-truth verbs and top-K model predictions and Macro-Averaged Verb Recall@K. For SRL we report CIDEr [33],

CIDEr-Vb: Macro-averaged across verbs, CIDEr-Arg: Macro-averaged across roles, LEA [20], and ROUGE-L [15]. For more details on the metrics pleas refer to [27].

**Implementation details.** We implement our model in Pytorch [22]. We extract event (video) features from a pretrained SlowFast model [8] for video representation (provided by [27]). For object features, we use a FasterRCNN model [24] provided by BUTD [2] pretrained on the Visual Genome dataset [13]. We sample frames at 1 fps from a 10 second video, resulting in $T = 11$ frames. We extract top $M = 15$ boxes from each frame, resulting in 165 objects per video.

All the three Transformers have the same configurations - they have 3 layers with 8 attention heads, and hidden dimension 1024. We use the tokenizer and vocabulary provided by VidSitu [27] which uses byte pair encoding. We have 3 types of learnable embeddings: (i) event position embeddings $\text{PE}_i$ with 5 positions corresponding to each event in time; (ii) object localization 2D spatial embedding; and (iii) role embeddings, for each of the 11 roles. The verb classification MLP has a single hidden layer of 2048 d and produces an output across all 1560 verbs. The role classification MLP also has a single hidden layer of 1024 d and produces output in a multi-label setup for all the 11 roles mentioned above. We threshold role prediction scores with $\theta_{\text{role}} = 0.5$.

We use the Adam optimizer [12] with a learning rate of $10^{-4}$ to train the whole model end-to-end. As we use pretrained features, we train our model on a single RTX-2080 GPU, batch size of 16.

## 4.1 Grounding SRL: Annotation and Evaluation

As free form captions and their evaluation can be ambiguous, we propose to simultaneously ground each correct role in the spatio-temporal domain. To evaluate grounding performance, we obtain annotations on the validation set. We select the same $T = 11$ frames that fed to our model sampled at 1fps. For each frame, we ask annotators to see if the visual roles (*agent*, *patient*, *instrument*), can be identified by drawing a bounding box around them using the CVAT tool [1] (see Appendix **??** for a thorough discussion). For each event $i$ and role $k$, we consider all valid boxes and create a dictionary of annotations $\mathcal{G}_{ik}$ with keys as frame number and value as bounding box. During prediction, for each role $r \in \hat{\mathcal{R}}_i$, we extract the highest scoring bounding box as in Eq. 11. The Intersection-over-Union (IoU) metric for an event consists of two terms. The first checks if the selected frame appears in the ground-truth dictionary, while the second compares if the predicted box has an overlap greater than $\theta$ with the ground-truth annotation,

$$\text{IoU@}\theta = \frac{1}{|\mathcal{R}_i|} \sum_{k=1}^{|\mathcal{R}_i|} \mathbb{1}[\hat{b}_t^* \in \mathcal{G}_{ik}] \cdot \mathbb{1}[\text{IoU}(\hat{b}_m^*, \mathcal{G}_{ik}[t]) > \theta]. \tag{15}$$

## 4.2 Grounded SRL Ablations

We analyze the impact of architecture choices, role query embeddings, and applying a mask in the cross-attention of the role-object decoder. All ablations in this section assume access to the ground-truth verb or roles as this allows us to analyze the effect of various design choices. Similar to [37] we observe large

Table 1: Architecture ablations. All the models use event-aware cross-attention. + indicates stages of the model. V: Video encoder, VO: Video-Object encoder, VOR: Video-Object-Role encoder, RV: Role-Video decoder, RO: Role-Object decoder, and C: Captioning Transformer.

| # | Architecture | Query Emb. | CIDEr | IoU@0.3 | IoU@0.5 |
|---|---|---|---|---|---|
| 1 | RV + C | Role + GT-verb | 47.91 ± 0.53 | - | - |
| 2 | RO + C | Role + GT-verb | **70.48** ± 1.09 | 0.14 ± 0.01 | 0.06 ± 0.003 |
| 3 | VOR + C | Role + Event | 67.4 ± 0.81 | 0.22 ± 0.00 | 0.09 ± 0.002 |
| 4 | V + RO + C | Role + Event | 69.15 ± 0.62 | 0.23 ± 0.03 | 0.09 ± 0.01 |
| 5 | VO + RO + C | Role + Event | 68.54 ± 0.48 | **0.29** ± 0.013 | **0.12** ± 0.01 |

variance across runs, therefore we report the average accuracy and the standard deviation over 3 runs for all the ablation experiments and 10 runs for the proposed model (VO+RO+C).

**Architecture design.** We present SRL and grounding results in Table 1. Rows 1 and 2 use a two-stage Transformer decoder (ignoring the bottom video-object encoder). As there is no event embedding $\mathbf{e}_i'$, role queries are augmented with ground-truth verb embedding. Using role-object pairs (RO) is critical for good performance on captioning as compared to role-video (RV), CIDEr 70.48 vs. 47.91. Moreover, using objects enables weakly-supervised grounding. Row 3 represents a simple

Transformer encoder that uses self-attention to model all the video events, objects, and roles (VOR) jointly. As before, role-object attention scores are used to predict grounding. Incorporating videos and objects together improves the grounding performance.

We switch from a two-stage to a three-stage model between rows 1, 2, 3 vs. 4 and 5. Rows 2 vs. 5 illustrates the impact of including the video-object encoder. We see a significant improvement in grounding performance 0.14 to 0.29 for IoU@0.3 and 0.06 to 0.12 for IoU@0.5 without significantly affecting captioning performance. Similarly, rows 4 vs. 5 demonstrate the impact of contextualizing object embeddings by events. In particular, using contextualised object representations $\mathbf{o}'_{mt}$ seems to help as compared against base features $\mathbf{x}^o_{mt}$.

**Role query embeddings design.** Prior works in situation recognition [7, 27, 35] use verb embeddings to identify entities from both images or videos. In this ablation, we show that instead of learning verb embeddings that only capture the uni-dimensional meaning of a

Table 2: Comparing role query embeddings.

| # | Query Emb. | CIDEr | IoU@0.3 | IoU@0.5 |
|---|---|---|---|---|
| 1 | Role only | $68.61 \pm 0.61$ | $0.27 \pm 0.011$ | $0.11 \pm 0.009$ |
| 2 | Role + GT-verb | $68.71 \pm 1.06$ | $0.25 \pm 0.02$ | $0.10 \pm 0.01$ |
| 3 | Role + Event | $68.54 \pm 0.48$ | $\mathbf{0.29} \pm 0.013$ | $\mathbf{0.12} \pm 0.01$ |

verb and ignore the entities involved, event (or video) embeddings remember details and are suitable for SRL. In fact, Table 2 (architecture: VO + RO + C) row 2 vs. 3 show that event embeddings are comparable and slightly better than GT-verb embeddings when evaluated on SRL and Grounding respectively, eliminating the need for GT verbs. Somewhat surprisingly, we see that the role embeddings alone perform quite well. We believe this may be due to role embeddings (i) capture the generic meaning like *agent* and *patient* and can generate the correct entities irrespective of the action information; and (ii) the role query attends to object features which are contextualised by video information, so the objects may carry some action information with them.

**Masked cross-Attention in RO decoder.** We use masking in event-aware cross-attention to ensure that the roles of an event attend to objects coming from the same event. As seen in Table 3 (model: VO + RO + C, query is role + event embedding), this reduces the object pool to search from and improves both the SRL and Grounding performance.

Table 3: Impact of masking in RO decoder.

| Mask | CIDEr | IoU@0.3 | IoU@0.5 |
|---|---|---|---|
| No | $67.02 \pm 0.51$ | $0.25 \pm 0.02$ | $0.10 \pm 0.012$ |
| Yes | $\mathbf{68.54} \pm 0.48$ | $\mathbf{0.29} \pm 0.013$ | $\mathbf{0.12} \pm 0.01$ |

## 4.3 SRL SoTA comparison

In Table 4, we compare our results against VidSitu [27] and a concurrent work that uses far better features [37]. We reproduce results for VidSitu [27] by teacher-forcing the ground-truth role pairs to make a fair comparison while results for work [37] are as reported in their paper. Nevertheless, we achieve state-of-the-art performance with a 22 points gain in CIDEr score over [27] and a 8 point gain over [37], while using features from [27]. Moreover, our model allows grounding, something not afforded by the previous approaches.

Table 4: SoTA comparison, results for SRL and grounding with GT verb and role pairs.

| Method | CIDEr | C-Vb | C-Arg | R-L | Lea | IoU@0.3 | IoU@0.5 |
|---|---|---|---|---|---|---|---|
| SlowFast+TxE+TxD [27] | 46.01 | 56.37 | 43.58 | 43.04 | **50.89** | - | - |
| Slow-D+TxE+TxD [37] | $60.34 \pm 0.75$ | $69.12 \pm 1.43$ | $53.87 \pm 0.97$ | $43.77 \pm 0.38$ | $46.77 \pm 0.61$ | - | - |
| VideoWhisperer (Ours) | $\mathbf{68.54} \pm 0.48$ | $\mathbf{77.48} \pm 1.52$ | $\mathbf{61.55} \pm 0.79$ | $\mathbf{45.70} \pm 0.30$ | $47.54 \pm 0.55$ | $\mathbf{0.29} \pm 0.013$ | $\mathbf{0.12} \pm 0.01$ |
| Human Level | 84.85 | 91.7 | 80.15 | 39.77 | 70.33 | - | - |

## 4.4 GVSR: Joint Prediction of Video Situations

The primary goal of our work is to enable joint prediction of the verb, roles, entities, and grounding.

**Verb prediction** is an extremely challenging problem due to the long-tail nature of the dataset. In Table 5, we evaluate verb prediction performance when training the model for verb prediction only (rows 1-3) or training it jointly for GVSR (rows 4, 5). Using a simple video-only transformer encoder boosts performance over independent predictions for the five event clips (46.8% to 48.8%, rows 1 vs. 2). Including objects through the video-object encoder (row 3) provides an additional boost resulting in the highest performance at 49.73% on Accuracy@1.

A similar improvement is observed in rows 4 to 5 (V vs. VO stage 1 encoder). Interestingly, the reduced performance of rows 4 and 5 as compared against rows 1-3 is primarily because the best epoch corresponding to the highest verb accuracy does not coincide with highest SRL performance. Hence, while the verb Accuracy@1 of the GVSR model does reach 49% during training it degrades subsequently due to overfitting. Nevertheless, we observe that the macro-averaged Recall@5 is highest for our model, indicating that our model focuses on all verbs rather than just the dominant classes. In Appendix **??**, we show the challenges of the large imbalance and perform experiments that indicate that classic re-weighting or re-sampling methods are unable to improve performance in a meaningful mannner. Addressing this aspect is left for future work.

Table 5: Verb prediction performance. Rows 1-3 train only for verb prediction. Rows 4, 5 are trained for GVSR.

| # | Architecture | Acc@1 | Acc@5 | Rec@5 |
|---|---|---|---|---|
| 1 | Baseline [27] | 46.79 | 75.90 | 23.38 |
| 2 | V | 48.82 | 78.01 | 23.32 |
| 3 | VO | **49.73** | **78.72** | 24.72 |
| 4 | V + RV + C | 40.83 | 70.73 | 24.37 |
| 5 | VO + RO + C | 45.06 | 75.59 | **25.25** |

**Understanding role prediction.** The verb-role prediction accuracy is crucial for GVSR, since the SRL task is modelled on role-queries. In Table 6 we analyse role prediction in various settings to understand its effect on SRL. Previous work [27] used ground-truth verbs for SRL, while roles and their entities or values are predicted sequentially. This setting is termed "GT, Pred" (row 2) as it uses the ground-truth verb but predicts the roles. We argue that as the verb-role mapping $\mathcal{P}$ is a deterministic lookup table, this setting is less interesting. We enforce a "GT, GT" setting with ground-truth verbs and roles in [27] by teacher-forcing the GT roles while unrolling role-noun predictions (row 1). Another setting is where the verb is predicted and roles are obtained via lookup, "Pred, GT map" (row 3). Note that this enables end-to-end SRL, albeit in two steps. The last setting, "Pred, Pred" predicts both verb and role on-the-fly (row 4).

Comparing within variants of [27], surprisingly, row 1 does not perform much better than row 2 on CIDEr. This may be because the model is trained on GT verbs and is able to predict most of the roles correctly (row 2, Role F1 = 0.88). Subsequently, both rows 3 and 4 show a large performance reduction indicating the over-reliance on ground-truth verb. We see similar trends for our models. Rows 7 and 8 with predicted verb-role pairs lead to reduced SRL performance as compared against rows 5 and 6. Nev-

Table 6: Role prediction in various settings. Role F1 is the F1 score averaged over all role classes.

| # | Architecture | Verb | Role | V. Acc@1 | Role F1 | CIDEr |
|---|---|---|---|---|---|---|
| 1 | VidSitu [27] | GT | GT | - | - | 46.01 |
| 2 | | GT | Pred | - | 0.88 | 45.52 |
| 3 | | Pred | GT map | 46.79 | - | 29.93 |
| 4 | | Pred | Pred | 46.79 | 0.47 | 30.33 |
| 5 | RO+C | GT | GT | - | - | 70.48 |
| 6 | VO+RO+C | Pred | GT | 45.06 | - | 68.54 |
| 7 | VO+RO+C | Pred | GT map | 45.06 | - | 51.24 |
| 8 | VO+RO+C | Pred | Pred | 44.05 | 0.44 | 52.30 |

ertheless, our "Pred, Pred" CIDEr score of 52.3 is still higher than the baseline "GT, GT" at 46.0. Appendix **??** discusses further challenges of multi-label and imbalance in predicting roles.

**GVSR.** We evaluate our end-to-end model for grounded video situation recognition. In order to enable end-to-end GVSR in [27], we use it in the "Pred, Pred" setting discussed above, that allows verb, role, and SRL predictions. Table 7 shows that our model improves SRL performance over Vidsitu [27] by a margin of 22% on CIDEr score. In addition to that, our model also enables Grounded SRL, not achievable in VidSitu [27].

Table 7: GVSR: Results for end-to-end situation recognition. Our model architecture is VO + RO + C.

| Model | Prediction | | | Verb Acc@1 | CIDEr | IoU | |
|---|---|---|---|---|---|---|---|
| | Verb | Role | SRL | | | 0.3 | 0.5 |
| VidSitu [27] | ✓ | ✓ | ✓ | 46.79 | 30.33 | - | - |
| VideoWhisperer | ✓ | ✓ | ✓ | 44.06 | 52.30 | 0.13 | 0.05 |
| | ✓ | GT | ✓ | 45.06 | 68.54 | 0.29 | 0.12 |

## 4.5 Qualitative Results and Limitations

We visualize the predictions of VideoWhisperer (Pred-GT) in Fig. 3 for one video of 10 seconds[1] and see that it performs reasonably well given the complexity of the task. VideoWhisperer correctly

---

[1]More examples on our project page, `https://zeeshank95.github.io/grvidsitu/GVSR.html`.

| Event | Frame 1 | Frame 2 | Frame 3 | Verb | Arg0 | Arg1 | Arg2 | ADir | AMnr | ALoc | AScn |
|---|---|---|---|---|---|---|---|---|---|---|---|
| Ev1 | | | | walk.01 | woman in white dress | | | towards the door | slowly | | in a house |
| | | | | walk.01 | woman wearing white | | | towards a door | slowly | | inside of a room with purple walls |
| Ev2 | | | | walk.01 | woman in white dress | herself | | around | | | in a house |
| | | | | turn.01 | woman wearing white | herself | | back | | | inside of a room with purple walls |
| Ev3 | | | | walk.01 | woman in white dress | herself | to get to the door | towards the door | | | in a house |
| | | | | reach.03 | woman wearing white | her arm | to open a cabinet | in front of her | | | inside of a room with purple walls |
| Ev4 | | | | open.01 | woman in white dress | door | | | quickly | | in a house |
| | | | | open.01 | woman wearing white | a cabinet | | | one at a time | | inside of a room with purple walls |
| Ev5 | | | | write.01 | woman in white dress | the door | | | | | in a house |
| | | | | rummage.01 | woman wearing white | shelves | | | | | inside of a room with purple walls |

Figure 3: We show the results for a 10s clip that can be viewed here: https://youtu.be/q6j_0vS_NNM?t=175. The video is broken down to 5 events indicated by the row labels Ev1 to Ev5. At a 1fps sampling rate, we obtain boxes from 3 frames for each event (with Frame3 of event $i-1$ being the same as Frame1 of event $i$). On the right side of the table, we show the predictions for the verb and various roles in the "Pred GT" mode, discussed in Table 6 (row 6). Predictions are depicted in blue, while the ground-truth is in green. Each role is assigned a specific color (see table header), and boxes for many of them can be found overlaid on the video frames (with the same edge color).

predicts verbs for actions like "open" and "walk". Given the large action space and complex scenes, there can be multiple correct actions, *e.g.* in Ev2 we see a reasonable "walk" instead of "turn".

For SRL, the model generates diverse captions with good accuracy, like "woman in white dress". Even though the ground-truth is syntactically different, "woman wearing white", they both mean the same. In fact, this is our primary motivation to introduce grounding. In Ev3, the model incorrectly predicts "walk" as the verb instead of "reach". While "walk" does not have the role Arg2, we are able to predict a valid caption "to get to the door" while grounding the woman's arm in Frame3. We see that our model correctly understands the meaning of Arg2 as we use ground-truth role embeddings combined with event features for SRL. This shows the importance of event embeddings, as they may recall fine-grained details about the original action even when there are errors in verb prediction.

For grounding SRL, we see that the model is able to localize the roles decently, without any bounding box supervision during training. While we evaluate grounding only for Arg0, Arg1, and Arg2 (when available), we show the predictions for other roles as well. In Fig. 3, the model is able to ground the visual roles Arg0 and Arg1 correctly. For non-visual roles like *"Manner"*, the model focuses its attention to the face, often relevant for most expressions and mannerisms.

**Limitations** for our current model are with verb and role prediction and disambiguation, improving the quality and diversity of captions to go beyond frequent words, and the division of attention towards multiple instances of the same object that appears throughout a video (details in Appendix **??**). Nevertheless, we hope that this work inspires the community to couple videos and their descriptions.

## 5 Conclusion

We proposed GVSR as a means for holistic video understanding combining situation recognition - recognizing salient actions, and their semantic role-noun pairs with grounding. We approached this challenging problem by proposing VideoWhisperer, that combines a video-object encoder for contextualised embeddings, video contextualised role query for better representing the roles without the need for ground-truth verbs and an event-aware cross-attention that helps identify the relevant nouns and ranks them to provide grounding. We achieved state-of-the art performance on the VidSitu benchmark with large gains, and also enabled grounding for roles in a weakly-supervised manner.

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
