# OpenReview forum: "Grounded Video Situation Recognition"
_NeurIPS.cc/2022/Conference — NeurIPS 2022 Accept_

### Official Review · Reviewer_iCYa · 2022-06-20

**Rating:** 7
**Confidence:** 3
**Soundness:** 3 good
**Presentation:** 1 poor
**Contribution:** 3 good

**Summary:**

This paper introduces a new task of performing structured situation recognition over time in videos, called Grounded Video Situation Recognition (GVSR). The idea is that for a new video, a model must identify which verbs are happening, identify their arguments (specifying who did what to whom), and then determine where in the video that event happened.

The authors create a dataset for this task built around VidSitu [6], also devising a set of metrics to handle multiple events happening in the same video (5 events per video, each video having 10 seconds). The authors collect extra annotations of the validation set to benchmark human performance.

The paper introduces a new model for this task called VideoWhisperer. Frames are sampled from a video, and a pretrained object detector is used to extract object embeddings from the resulting frames. A separate pretrained video backbone extracts features from the whole video. A transformer encoder fuses these features together, and a separate transformer decoder is used to perform structured prediction (for the semantic role labeling over videos). Results and ablations show that this model performs well at both localized video situation recognition, as well as the non-localized case (VidSitu).

**Questions:**

* Qualitatively, what examples does this approach work well on and where does it struggle?
* How important are the choice of video representations (\phi_{vid} vs. \phi_{obj}) ?

**Limitations:**

There isn't a limitations section or a discussion on the societal impacts, adding this would improve the paper.

**Strengths And Weaknesses:**

To this reviewer, this paper is a strong overall contribution. It tackles an important and unsolved problem (fine-grained video understanding) and introduces a new dataset (Grounded VidSitu) on top of VidSitu that might help evaluate this. The proposed model seems like a first step in this direction (pipelining the problem of structured video understanding), and helpful ablations are provided to show where the model might be improved in future work.

Weaknesses:
* To this reviewer, the model feels a bit hacky and overly complex. I have a suspicion that a simpler model would perform better, but I'm also OK with this being a problem that future work ought to address.
* Adding a limitations/social impact section would improve the paper considerably.
* Grounded seems like an overloaded term, do you mean something like "Localized Video Situation Recognition"?
* The modeling section (Sec 3) was confusing to this reviewer. It might be helpful to discuss the high-level strategy first before jumping into the details, as well as the function of the ROTD/VOTE networks. Those acronyms like they are referred to that often, so maybe dropping them would boost understanding.
* Some qualitative examples and discussion would help.

---

> ### Author Response · Authors · 2022-08-02
> **Thanks iCYa and answers below.**
>
> We thank the reviewer for their positive feedback. We especially appreciate the detailed summary that captures the salient features of our work. We address the raised weaknesses and questions below.
>
> **1. Model feels a bit hacky and complex**
>
> We request the reviewer to refer to the answer about the model’s complexity in the general comment to all reviewers. We believe that our model captures the three stages of the structured prediction task quite well, and each stage is an important component of the model (as shown via architecture ablations in Table 1). We also agree with the reviewer that while a simpler model may exist, developing it could be future work, something we will eagerly watch for.
>
> **2. Missing limitations section**
>
> Due to lack of space, we added the limitations section to the Appendix C of the supplementary material. We will move this to the main paper in the updated version. We discuss three main challenges in GVSR, including role disambiguation, generation of rich and descriptive captions, and the challenges to identify the right object from a large pool.
>
> **3. Grounded seems overloaded**
>
> To our understanding, “grounding” is used when an object class or a text query (e.g., referring expression) is localized in a given image/video. In our case, we think of role queries from Eq. 6 as a form of text query that asks a targeted question about a specific verb-role pair. For example, “*who* is *jumping*?”, where *who* comes from $r_k$ and *jumping* from $e_i’$. Our model performs both: localizes the entity in the spatio-temporal video with weak supervision and generates a natural language description for the entity using the caption decoder. A second reason to use Grounded is inspiration from Grounded Situation Recognition [13] that also needs to predict the noun label (entity) corresponding to the verb-role pair while also localizing it in the image. However, we are open to discuss this further with the reviewers.
>
> **4. Improvements to Sec. 3, removing acronyms**
>
> Thank you for the suggestions. We will certainly discuss the high-level goals of the model before diving into the details by including a paragraph similar to the general comment to all reviewers that connects the three predicted outputs with the three stages of the model. We also thank the reviewer for pointing out the unnecessary acronyms, VOTE and ROTD. We indeed prefer to simply call them encoder and decoder and will remove them to improve readability.
>
> **5. Qualitative examples, where does the model struggle?**
>
> Due to lack of space, we included detailed qualitative results with verb prediction, grounding visualization, and SRL captions as part of the supplementary material. We discussed the results depicted in Fig. 3 in Appendix A of the supplementary material. One figure showing the results is insufficient - hence we also included a template based and self-contained HTML file (GVSR.html) that shows predictions of all 5 events for a total of 10 videos. As mentioned a few paragraphs above, the qualitative results expose some limitations of our model, which we reason about in Appendix C of the supplementary material.
>
> **6. Importance of choice of video representations ($\phi_{vid}$ vs. $\phi_{obj}$)**
>
> Sorry, but we did not understand the question completely. Is the reviewer asking about the (A) impact of video vs. object features, or (B) impact of differences in video representations?
>
> We believe that (A) is best answered through our ablation study on architectures (Sec. 4.3, Architecture Design, Table 1). When using the ground-truth verb embedding in rows 1 and 2, we see that object features drastically improve captioning performance, while also enabling grounding. Rows 3 and 4 add the missing stage1 to the model to encode video and object representations. Here as well, we see that the video-object encoder (row 4) outperforms the video-only encoder (row 3) on localization of verb-role entities. In addition to the SRL evaluation, Table 5 analyzes verb prediction performance and we see that the video-object encoder (row 3) achieves highest verb prediction accuracy.
>
> In case the question is related to (B), we request the reviewer to look at the original paper VidSitu [6]. In particular, Table 5 of [6] shows the impact of different backbones and Kinetics pretraining on verb prediction, SlowFast with Kinetics pretraining outperforms other variants. In our work, we use the extracted features corresponding to the best model shared by [6] directly. Using precomputed object and video features results in heavy computational savings, and we are able to train our model on a single RTX-2080 Ti GPU (12 GB RAM) with a batch size of 16.

---

> > ### Comment · Reviewer_iCYa · 2022-08-08
> > **thanks for the response, still recommend acceptance**
> >
> > Thanks for the response!
> >
> > 1./4. Thanks for agreeing to improve the readability of Sec3! That could also address my concern around the complexity of the model.
> >
> > 2. Thanks!
> >
> > 3. I still think this definition of "grounded" is a bit overloaded. Since you used "localized" to define your version of grounding, it might be better just to describe this work as "Localized Video Situation Recognition" IMO.
> >
> > 5. Thanks! Incorporating the html file into the paper (or putting it online somewhere) would be great upon acceptance.
> >
> > 6. I was asking more about A. Good point that this could be answered by the ablation study, it might be good to make that point more explicit in the paper.

---

> > > ### Author Response · Authors · 2022-08-09
> > > **Thanks!**
> > >
> > > We thank the reviewer for their response. We will certainly incorporate the valuable feedback given by the reviewer, especially on readability improvements to Section 3.

---

### Official Review · Reviewer_tM2B · 2022-07-11

**Rating:** 6
**Confidence:** 3
**Soundness:** 3 good
**Presentation:** 3 good
**Contribution:** 3 good

**Summary:**

VidSitu involves predicting the actions, entities involved in the actions, and simple relations between the actions, given a short video clip. This paper proposes spatio-temporal grounding (GVSR — Grounded Video Situation Recognition) as an addition to the VidSitu task. Spatio-temporal grounding involves localizing the entities which helps disambiguation of entities and co-referencing entities across different actions.

The paper also proposes a novel Transformer model that (unlike previous works) jointly predicts all task outputs — verbs, verb-role pairs, nouns, grounding. The VideoWhisperer model has 3 stages:

1. Learning contextualized features for the video and key objects simultaneously. Predicts the action label (verb roles) for each event.
2. Semantic role labeling, i.e., generating a caption (noun) for each verb role in each event.
3. Localization is achieved by finding the largest attention score for a bounding box in the (ROTD) Transformer, for each verb-role in each event.

The proposed model has a few advantages — (1) results are predicted jointly, in a single forward pass, (2) improves entity captioning accuracy and (3) localizes verbs accurately despite not having access to grounding annotations during training.


**Questions:**


-  The main challenge with verb prediction seems to be the long-tailed distribution of verbs in VidSitu. Have the authors considered simple approaches to handle this, e.g., class-balanced loss [a] and/or focal loss [b] ? While these are unlikely to completely solve the verb imbalance problem, they may result in some improvements.
- The object features are only computed from subsampled frames from the video. However, the event embeddings are computed from video features from the Slow-Fast network. I wonder if the authors have tried to compute event embeddings from subsampled frame features instead of full video. This would reduce the computational overhead (video vs. keyframes) if true.

References
[a] Cui, Yin, et al. "Class-balanced loss based on effective number of samples." CVPR. 2019.
[b] Lin, Tsung-Yi, et al. "Focal loss for dense object detection." CVPR. 2017.


**Limitations:**

This work contributes to the general direction of understanding semantics in videos. However, current learning methods require ground-truth labels and large amounts of data. A huge challenge in this area is the difficulty in scaling this up to a large number of action labels (verbs) and roles.

**Strengths And Weaknesses:**

Grounding semantic predictions in the given video, is an important problem that is well-motivated. Accurate grounding can improve the reliability, robustness, trust in the model, and possibly also prediction accuracy.

The GVSR task is an obvious extension of GSR [13] (which focused on grounding in imSitu [7]) to VidSitu [6]. Although the originality of the task is low, it is indeed useful, and improves the experimental setup of the existing VidSitu task.

The main contribution of the paper is (a) localization annotations (for testing) on VidSitu, and (b) the VideoWhisperer model. Unlike previous works on VidSitu, the proposed model is trained end-to-end, and outputs the entire set of structured outputs in a single forward-pass. This architecture is a significantly simpler implementation. Such an architecture allows learning (common) representations, and modeling the context across sub-tasks of VidSitu.

By leveraging cross-attention across object bounding box regions in the transformer model, localization can be achieved by weak supervision (via the caption loss). There is no need for ground-truth localization annotations. This allows one to potentially scale up the training as long as only language annotations are available.

VidSitu requires the prediction of the action label (verb), the entities (nouns) associated with the verb-role and the relationship between events. The ground-truth language descriptions of the entities may be ambiguous, and have some variance — e.g., “Woman in blue shirt”, “woman in glasses”, etc. could all refer to the same person. Thus, evaluating captions is challenging, and imprecise. The entity localization task (measured by IoU) is less ambiguous. Further, it provides an additional dimension of understanding and interpretability of the model predictions. Thus GVSR alleviates some of the evaluation issues in VidSitu (to an extent).

There are significant improvements in performance in entity-captioning accuracy + localization of verb-roles.

---

> ### Author Response · Authors · 2022-08-02
> **Thanks tM2B and answers below.**
>
> We thank the reviewer for an overall positive feedback and thoughtful interpretation on how localization can overcome ambiguous evaluation. Below we address the concerns and also include some experiments on verb prediction using several methods that are commonly reported to address long-tailed distributions.
>
> **1. GVSR is an obvious extension of GSR [13]**
>
> Grounded Situation Recognition (GSR) can be thought of as an extension, grounded version of imSitu. However, doing the same for GVSR is not straightforward since GSR considers localization in a single image, while in GVSR, we may need to localize roles in events that do not even appear in the same event (e.g., when two people are talking, person A is localized in the current event, while person B could be localized in the previous/future event). Hence, our localization annotations are not restricted to within the event and consider all 5 events (entire 10s clip) together. Additionally, we feel that it isn’t obvious how localization should be performed across multiple movie shots, where the same character may re-appear. We resolve the temporal ambiguity and computational complexity by sub-sampling frames, but picking them such that there is shared information across events due to the shared borders (e.g., the frame at 2s belongs to both events 0-2s and 2-4s). We think this is a non-trivial extension.
>
> **2. Long-tailed distribution of verbs**
>
> The VidSitu dataset encompasses a large number of verbs and has a long-tailed distribution. In fact, the number of verbs, 1560, is 2-4x larger than popular large-scale video action recognition datasets (Kinetics400 / Kinetics700).  We believe that these are the two key challenges that result in lower performance.
>
> As suggested, we experiment with three common ideas to handle long-tailed distributions. (i) Loss re-weighting applies weights corresponding to the inverse verb frequency to the cross-entropy loss; (ii) Focal loss is applied as described in [A] (with gamma = 2.0); and in (iii) Balanced sampling, we apply a weight for each sample such that the dataloader picks samples with a higher weight.
>
> ```
> | Method                | Verb Acc@1 |
> | --------------------- | --------–- |
> | Baseline [6]          | 46.79      |
> | V (in submission)     | 48.82      |
> | V (loss re-weighting) | 48.91      |
> | V (Focal loss)        | 47.81      |
> | V (balanced sampling) | 35.38      |
> ```
>
> Unfortunately, we do not see any significant improvement using these simple approaches. We have observed that the dataset is very challenging and has complex movie events with fast shot changes and many actions can be confusing. For example in Figure 3 (supplementary material) the woman turns while walking, but the model predicts “Walk” instead of “Turn” which is the dominant, but less significant action (if one considers duration). Balanced sampling in particular leads to a significant drop since our sample consists of 5 event clips, each with a verb. When rare verbs are oversampled, co-occurring events with potentially not-so-rare verbs are also oversampled, leading to a skewed training dataset. (This is similar to the challenges of applying balanced sampling to multi-label classification.)
>
> **3. Event features from keyframes**
>
> Thanks for this suggestion, it is indeed interesting to use sub-sampled frames for event embeddings to decrease the computational complexity. We suspect models like Temporal Relational Reasoning (TRN) [A] may work well. However, we will need to pre-train the network first on a large scale dataset like Kinetics and then fine-tune it on VidSitu verbs to adapt it to movie clips, a computationally intensive exercise. Instead, we reduce computation by re-using pre-trained models to extract object (FasterRCNN [35] provided by BUTD [2]) and video features (SlowFast provided by VidSitu [6]).
>
> References:
>
> [A] Lin, et al. “Focal Loss for Dense Object Detection.” ICCV, 2017.
>
> [B] Zhou, et al. "Temporal relational reasoning in videos." ECCV, 2018.

---

> > ### Comment · Reviewer_tM2B · 2022-08-08
> > **Thanks for the authors' response**
> >
> > 1. Thanks for the additional context. I completely agree that the questions of how to define and approach the GVSR task, are non-trivial. I reiterate that while the \emph{idea} of the GVSR task might be obvious after GSR, the task is no less important because of it. Indeed, the task is obvious because it is a logical and important direction of research in video understanding. In my view, this is not a weakness; it is a strength. There is no necessity that every proposed task be surprisingly new.
> >
> > 2. Thanks for the experiments. It looks like handling the long-tailed verb distribution is an important problem to solve to push performance further in GVSR.
> >
> > The additional explanation regarding oversampling is insightful. It brings up the question of how can one perform balanced sampling of (rare) parts of structured data (without oversampling co-occurring frequent data).
> >
> > I also agree on the other point -- that shot changes and complexity of movie events are major contributors to the challenging nature of the task.

---

> > > ### Author Response · Authors · 2022-08-09
> > > **Thanks!**
> > >
> > > We thank the reviewer for their response and apologize for the misunderstanding. We are glad to see that the reviewer acknowledges the non-trivial nature of the problem definition. We agree with the reviewer that addressing the long-tail in a challenging setup such as GVSR is an important aspect of improving performance and hope to see the community build exciting future works in this direction.

---

### Official Review · Reviewer_4WkR · 2022-07-12

**Rating:** 5
**Confidence:** 3
**Soundness:** 2 fair
**Presentation:** 2 fair
**Contribution:** 2 fair

**Summary:**

This paper studied the problem of video situation recognition, which predicts structured information like multiple events, relationships, actions, and verb-robe pairs.  The authors propose a three-stage Transformer model, VideoWhisperer to handle the task. The model first learns contextualized embeddings for video features in parallel with key objects that appear in the video clips. The model then attends and pools information from object embeddings to localize answers to questions. The final stage of the model generates answers as captions to describe each verb-role pair in the video. Experiments are conducted on the VidSitu dataset, achieving better performance than the baseline models.

**Questions:**

Please see the **weaknesses** in the previous section.

**Limitations:**

Yes, the authors adequately addressed the limitations and potential negative social impact of their work.

**Strengths And Weaknesses:**

**Strengths**
1. The paper overall is well written and easy to follow.
2. The idea to decompose the task into different stages seems to be technically sound and the ablation study in the experimental section shows its effectiveness.
3. As the conclusion of the introduction, the authors also provide new data annotation to an existing dataset.

**Weaknesses**
1. The first concern is about the novelty of the main idea of the paper. The proposed mode seems to be complicated to decompose the task into three stages. Whether it is possible to compare the computation complexity between the proposed method and the baseline.
2. Second, as a method paper, this paper only conducts experiments on a recent proposed dataset. And the dataset contains only a simple baseline. This makes the paper less convincing. Whether it is possible to extend the proposed method to other video situation understanding datasets like (1) to further prove the effectiveness of the proposed method?
3. It will also be great to implement more baselines on the existing benchmark.

 (1). Wu, Bo, et al. "STAR: A benchmark for situated reasoning in real-world videos." Thirty-fifth Conference on Neural Information Processing Systems Datasets and Benchmarks Track 2021.

---

> ### Author Response · Authors · 2022-08-02
> **Thanks 4WkR and answers below.**
>
> We thank the reviewer for their positive feedback and suggestions for other datasets and baselines. Below we address concerns one by one.
>
> **1. The model is complicated, complexity comparison to baseline**
>
> We request the reviewer to please refer to the detailed answer about the model’s complexity in the general comment to all reviewers. In summary, (i) we think that the design of our model is necessary to predict several fine-grained aspects of the GVSR task; (ii) our model can easily be trained on a single 12GB GPU and predicts verbs and roles at stage 1,  localizes entities in stage 2, and captions them in stage 3; and (iii) inference for our model is more efficient than the baseline [6] as we are able to generate captions in parallel, leading to a reduction in wall-clock time from (on average) 0.94s for [6] to 0.4s for our model.
>
> **2. Extending the method to the STAR benchmark**
>
> Thanks to the reviewer for pointing us to the STAR benchmark [A]. While we would be happy to evaluate our approach on another dataset, we think VidSitu [6] is the most appropriate dataset for this task. In fact, the contribution of our work is not only a model, but an extension of VidSitu with localization annotations that enables end-to-end and *Grounded* Video Situation Recognition. Below, we highlight the key differences between GVSR and STAR:
>
> - Difference in tasks: *Situated reasoning* in STAR is defined based on human-object interactions and relationships in time with a set of actions and objects. The task is evaluated through multiple-choice QA. Situations in GVSR are defined based on Semantic Role Labeling, and have a fixed hierarchical linguistic structure used to describe *situations* with verb-role pairs and their entities. GVSR requires structured prediction of labels, the action labels are an order of magnitude higher (1560 vs. 111) and the entities are described with natural language (instead of fixed labels).
>
> - Different inputs and outputs: The input data for STAR is a video, a natural language question about a situation, and a set of multiple choice answers. The task requires using the question to attend and extract information from the visual cues to select the correct answer. The input to GVSR is a single video consisting of 5 events. Models are expected to predict per-event actions, localize verb-specific role entities in the video, and generate natural language descriptions for entities. The three stages of our model are specially designed to make these predictions and decode the entire situation with fine-grained details.
>
> To the best of our understanding, adapting our model to STAR will require significant changes to the architecture, and including logical reasoning as [A] seems to advocate will correspond to solving a different problem.
>
> **3. Additional baselines**
>
> VidSitu [6] is a recent dataset published in CVPR 2021, but the original authors presented extensive baselines for semantic role prediction (entity captioning) from GPT2 to trained Transformer encoder-only, decoder-only, or encoder-decoder models. They also show the effects of using different video features such as I3D or SlowFast, both on verb prediction and entity captioning. Secondly, [9] was published at CVPR 2022 (after the NeurIPS submission deadline), and proposes a video representation learning approach using contrastive learning and masked prediction. The obtained features push the captioning metric (CIDEr score) from 46.5 to 60.34. We compare our approach against the best models from both works in Table 4 and see that VideoWhisperer achieves an impressive 22 point improvement over [6] and an 8 point improvement over [9] while using the features provided by [6]. We expect that using the features from [9] will further improve performance, but were unable to obtain them.
>
> As fine-grained video understanding in the form of Semantic Role Labeling is a new and challenging problem, there aren’t many previous methods to compare against. In addition, we are also proposing an end-to-end solution GVSR. along with localization - making this a new task. In such a scenario, we think that some of our ablations can be considered as baselines. For example, in Table 1, we show the impact of discarding the stage 1 encoder (rows 1, 2 vs. 3, 4); including object representations for captioning (rows 1 vs 2); and also including object representations in the stage 1 encoder for improving the contextualized embeddings (row 3 vs. 4). We would be happy to consider other approaches that can be used as baselines without significant modifications.
>
>
> References:
>
> [A] Wu, et al. "STAR: A benchmark for situated reasoning in real-world videos."  NeurIPS Datasets and Benchmarks Track, 2021.

---

### Official Review · Reviewer_1XR4 · 2022-07-18

**Rating:** 7
**Confidence:** 3
**Soundness:** 3 good
**Presentation:** 3 good
**Contribution:** 3 good

**Summary:**

- This paper introduces a Transformer-based architecture for video situation recognition that a) augments prior work on Semantic Role Labeling in videos with spatiotemporal grounding, and b) removes the requirement of ground-truth role-verb mapping, enabling end-to-end Grounded Video Situation Recognition (GVSR).
- The proposed model -- VideoWhisperer -- has 3 main components -- a) a Transformer encoder to learn object embeddings per frame, and event embeddings, both contextually enriched via self-attention; b) a Transformer decoder to predict the entities for each role, conditioned on the object and event embeddings; c) a Transformer decoder to autoregressively predict the caption for each role conditioned on the output features of (b).
- To evaluate the model on grounding, the authors collected annotations by asking users to draw bounding boxes around roles in each event clip, for the validation set. These annotations will be released and/or incorporated in the official evaluation toolkit. These annotations are not used for training (and so the proposed model is only weakly-supervised).
- Experiments are thorough, both the ablations to demonstrate the importance of each design choice, as well as comparison to prior work on the SRL benchmark. On the SRL benchmark, the proposed approach significantly outperforms prior work, and in addition, supports spatiotemporal localization, unlike prior works.

**Questions:**

- Which pretrained object detector backbone does VideoWhisperer use?
- It seems like role prediction accuracy (~45%) is limiting downstream performance to quite an extent. Some discussion on next steps to further improve this would be useful to include in the paper.

**Limitations:**

- The authors have included a brief overview of limitations in the appendix.

**Strengths And Weaknesses:**

The proposed model makes sense, is well-validated by experiments, and importantly enables joint prediction of actions, roles, spatiotemporal localization of roles, and captions (unlike prior approaches that need ground-truth role-verb annotations at test time). Great work!

---

> ### Author Response · Authors · 2022-08-02
> **Thanks 1XR4 and answers below**
>
> We thank the reviewer for their detailed review and positive feedback for our work. We are happy to see that the reviewer finds joint prediction of actions, roles, spatiotemporal localization of roles, and captions as an important contribution. We answer questions below.
>
> **1. Pretrained object detector**
>
> We use the popular FasterRCNN [35] model provided by Bottom-Up Top-Down Attention [2] that is pretrained on the Visual Genome dataset. This model has been widely adopted as it is trained on a large number of classes (more than COCO) and fine-grained object attributes. We mentioned this briefly in the implementation details, L253.
>
> **2. Improving role prediction accuracy**
>
> The reviewer is correct in their assessment. Role prediction directly affects the Situation Role Labeling (SRL) performance -- if the correct role is missed the contribution towards the metric for that role is considered to be 0, drastically lowering the SRL performance. In our work, role accuracy depends on the event (video clip) features, and both the verb and role losses affect the event embeddings. The severe imbalance in the roles leads to large fluctuations in role prediction performance (Appendix B.3 in the supplementary material mentions some details). We believe that better event representations can help improve performance and future works can look into video backbones specifically trained/designed for challenging movie scenes. We also think that addressing the long-tailed nature of the dataset is important.

---

### Author Response · Authors · 2022-08-02
**Thanks to all reviewers for their positive response and providing valuable feedback. We cite a few comments and also address a common concern from some reviewers.**

We are happy to see that reviewers are positive about several aspects of our work: enabling joint prediction of actions, roles, spatiotemporal localization of roles, and generation of captions (Reviewers 1XR4, tM2B), addressing fine-grained video understanding through localization (Reviewer iCYa), effective and well-validated experiments showing improved captioning accuracy (Reviewers 1XR4, 4WkR, tM2B, iCYa), and that the paper is well written and easy to follow (Reviewer 4WkR). We especially thank Reviewer 1XR4 for their encouragement on proposing a joint modeling task, “Great work!”, and Reviewer tM2B for their thoughtful interpretation of how language descriptions can be ambiguous through the wonderful example, “Woman in blue shirt or woman in glasses can refer to the same person”. In fact, this was our key motivating factor for introducing localization, as a less ambiguous evaluation that also provides “an additional dimension of understanding and interpretability”.

While we provide individual responses for each reviewer, we would also like to address a common concern shared by some reviewers related to the model’s complexity.

**The model looks complicated**
(Reviewers 4WkR, iCYa)

Video Situation Recognition (and GVSR) is built on the hierarchical linguistic structure from the theory of Semantic Role Labeling [8] which naturally decomposes the task into three stages. GVSR is an example of fine-grained video understanding that requires reasoning about multiple components of the video: (i) recognizing the key action verbs and their corresponding roles, (ii) localizing the entities corresponding to the role for the specific verb, and (iii) generating a natural language description for each entity. Our 3-stage model follows a straightforward approach to generate predictions for all 3 tasks. The first stage encoder predicts verbs and associated roles using object and event embeddings, the second stage decoder identifies entities relevant to the specific role, while the final third stage generates the semantic role labels or captions.

*Architecture.*
As we use pretrained models to extract event and object features, the model is in fact quite light weight and we are able to train it with a batch size of 16 on a single RTX-2080 Ti GPU (12 GB memory). Each stage consists of 3 Transformer layers, resulting in a total of 9 blocks. The complexity of self-attention typically depends on the number of input tokens, which is limited to $|\mathcal{B}| + |\mathcal{E}|$ (15*11 objects + 5 events). The cross-attention layers have no more than 30 query tokens (corresponding to at most 6 roles for each of the 5 events), and attend to the relevant boxes in that event through masking (3 frames, 15 objects per frame). The captioning stage is also limited to a max sentence of 15 words.

*Wall-clock time.*
By introducing a role decoder, we mix role information prior to captioning and show that captions can be generated in parallel for all roles of all events, while baselines [6, 9] generate captions sequentially, i.e. for a given event, the caption for role_i is decoded only after the caption of role_{i-1}. This results in an efficient wall-clock time for VideoWhisperer, with ~0.4s for inference on a 10s video (5 * 2s events), while the baseline [6] requires ~0.94 seconds. We will include this analysis in the updated version.

---

### Meta-Review · Area_Chair_5kwa · 2022-08-25

**Recommendation:** Accept
**Confidence:** Certain

**Metareview:**

All four reviewers agreed that this paper tackles an important problem, and the proposed approach is novel and well supported by sufficient empirical evidence. They also acknowledged the new data annotation contribution and that that the paper is generally written well. There were several nice suggestions to improve the paper. The authors are strongly encouraged to incorporate them in the final version.

**Award:**

No

---

### Decision · Program_Chairs · 2022-09-14

Accept